# Modular Compilation for a Hybrid Non-Causal Modelling Language

**Guerric Chupin** *,†  and **Henrik Nilsson** †

School of Computer Science, University of Nottingham, Nottingham NG8 1BB, UK; Henrik.Nilsson@nottingham.ac.uk
* Correspondence: Guerric.Chupin@nottingham.ac.uk
† These authors contributed equally to this work.

**Abstract:** Non-causal modelling is a powerful approach to modelling physical systems in a variety of domains from science and engineering. Non-causal modelling languages enable a high-level and modular approach to modelling. However, it is hard to compile non-causal languages modularly (in the sense of separate compilation). This causes difficulties when simulating large models for which code generation takes a long time, or structurally singular models in which parts of the model are allowed to change at runtime. In this work, we introduce a technique we call order-parametric differentiation to allow truly modular compilation. The idea is to generate (machine) code that can compute derivatives of any order of an expression as needed, thus allowing for ahead-of-time modular compilation of a hybrid non-causal language. We also develop a compilation scheme that enables using partial models as first-class objects in a seamless way and simulating them without the need for just-in-time compilation, even in the presence of structural dynamism. We present a performance evaluation of the scheme we used and study its shortcomings and possible improvements, demonstrating that it is a feasible complement to existing implementation techniques for cases where true modular compilation is a primary objective.

**Keywords:** non-causal modelling; modular compilation; automatic differentiation; order-parametric differentiation





## 1. Introduction

Non-causal modelling is a powerful approach to modelling systems described by differential equations. Its appeal resides in the ability to express models in terms of undirected equations; that is, without any a priori assumption that some of the variables are "known" and used for computing the "unknowns". This makes models highly modular—a model is written once and reused as needed; the compiler is in charge of determining which equation to use to compute which variable depending on the context. Modelling languages implementing this paradigm include Modelica [1] and Dymola [2]. By contrast, causal modelling languages, such as Simulink [3], Ptolemy [4] or Zélus [5], only allow the use of directed equations: the unknown on the left-hand side is computed by the expression on the right-hand side. This means that the model must be rewritten every time the causality changes.

However, from a compiler implementor's perspective, non-causal models are not modular in the slightest. A system of undirected equations form a Differential Algebraic Equation (DAE) whose simulation poses inherent difficulties, particularly the differentiation index [6]. In general, a non-causal model features constraint equations that restrict the set of possible solutions but cannot readily be used by a numerical solver in computing a solution. Recovering a system suitable for simulation can be done by differentiating some of the equations of the system. The resulting latent equations can then be used for simulation. The differentiation index is the number of times all or part of the a DAE has to be differentiated to recover an Ordinary Differential Equation (ODE), which corresponds

to a causal equation. In general, DAE solvers are only able to solve index-1 systems, or implicit ODE.

There exist efficient algorithms [7,8] for determining the set of equations to differentiate to obtain a system suitable for simulation. However, they all require a complete view of the model being simulated. Thus, before a model is completely assembled, it is not possible to fully determine the equations that must be differentiated for simulation. This is problematic for code generation: because the index of a DAE can be arbitrarily large [9], it is not possible to generate separate code ahead of time for all derivatives that may be needed. Hybrid modelling languages exacerbate the problem. Here, the equations of a model are allowed to change during simulation, meaning that equations have to be differentiated and new code generated as the simulation progresses. An example of this from the field of electronics is simulation of ideal diodes, where the number of modes is two to the power of the number of diodes, quickly making it impractical to generate code for all possible modes ahead of time [10] and thus necessitating an "on demand" approach to mode enumeration.

For the reasons above, hybrid non-causal languages have traditionally been either interpreted [11] or just-in-time (JIT) compiled [12,13]. Although recent work [14] has been proposed to compute the set of latent equation for a dynamic model, the model still has to be complete at the moment the algorithm is applied. In particular, all the possible set of equations the model can enter must be known. This rules out a certain category of hybrid languages [15] where the modeller can programmatically generate new equations during the simulation, making the number of modes potentially infinite and in general unknown before the simulation.

In this paper, to address these problems, we explore generating code capable of computing an arbitrary derivative of a formula. Computation of arbitrary derivatives of an expression has been explored in the automatic differentiation literature. For instance, Karczmarczuk [16] showed how, exploiting lazy evaluation, derivatives of arbitrary order can be computed on demand. However, questions of efficiency aside, the runtime support needed for lazy evaluation is considerable and not a particularly natural fit with the DAE solvers and other components typically used for implementing this class of languages. Instead, we consider an approach we call order-parametric differentiation—the code generated for an expression is parameterised on the order of the derivative. It can thus be used to compute any derivative at any point (including the undifferentiated value of the expression for order 0).

Our objective is to generate modular code in the usual, programming-language sense of the term. Compilers for programming languages, like C, Java or Haskell, compile the code for a function once and then simply use a symbol to jump into the body of the function when it is being called. This allows separately compiled modules to be joined by a linker, without further compilation as such. We want simulation code for models to be compiled into a function and use the same mechanism when one model is used in another, allowing separately compiled models to be linked in the conventional sense. We explore this in the context of our non-causal modelling language Hydra$_2$, a non-causal modelling language supporting structural dynamism. However, note that the key ideas presented are applicable generally and not limited to Hydra$_2$. Further, we would like to emphasize that there are pros and cons of our proposed approach, and as such we view it as a complement to existing implementation techniques for cases where true modular compilation is particularly important, not as a replacement for existing techniques.

The rest of the paper is organised as follows—in the next section, we present an introduction to non-causal hybrid modelling through the use of Hydra$_2$. We describe the usual steps needed to simulate a non-causal model and explain how this complicates ahead-of-time code generation. We then introduce the mathematical formalism we use for order-parametric differentiation. Next, we present performance benchmarks, comparing with a more traditional approach to the generation of simulation code. Finally we discuss future work addressing current limitations and present conclusions.

## 2. Non-Causal Hybrid Modelling

This section aims to provide an introduction to non-causal hybrid modelling using simple examples from electronics. This is done in the context of Functional Hybrid Modelling (FHM), as implemented in the language Hydra$_2$ (https://gitlab.com/chupin/hydra-v2, accessed on 29 March 2021). We start by introducing the fundamental concepts around which FHM is built and then show how it forms an expressive and modular modelling paradigm. The abstract syntax of Hydra$_2$ is given in Figure 1.

$\langle \tau_s \rangle ::= \alpha_s \mid \mathbb{R} \mid \langle \tau_s \rangle \; '\times' \; \langle \tau_s \rangle$

$\langle \tau_e \rangle ::= \alpha_e \mid \langle \tau_s \rangle \mid \langle \tau_e \rangle \; '\times' \; \langle \tau_e \rangle \mid \langle \tau_e \rangle \; '\to' \; \langle \tau_e \rangle \mid 'SR' \; \langle \tau_s \rangle$

$\langle unop \rangle ::= 'sin' \mid 'cos' \mid 'sqrt' \mid \ldots$

$\langle binop \rangle ::= '+' \mid '-' \mid '*' \mid '/' \mid '\hat{}'$

$\begin{aligned} \langle signal \rangle ::= \; & \langle expr \rangle \\ \mid \; & \langle identifier \rangle \\ \mid \; & 'der' \; \langle signal \rangle \\ \mid \; & \langle signal \rangle \; \langle binop \rangle \; \langle signal \rangle \\ \mid \; & \langle unop \rangle \; \langle signal \rangle \end{aligned}$

$\langle condition \rangle ::= 'when' \; \langle event \rangle \; '\text{->}' \; \langle ident \rangle \; '(' \; \langle sig \rangle^* \; ')'$

$\langle branch \rangle ::= 'mode' \; \langle ident \rangle \; '(' \; \langle pattern \rangle^* \; ')' \; '\text{->}' \; \langle relation \rangle^* \; \langle condition \rangle^*$

$\begin{aligned} \langle relation \rangle ::= \; & \langle signal \rangle \; '=' \; \langle signal \rangle \\ \mid \; & 'init' \; \langle signal \rangle \; '=' \; \langle signal \rangle \\ \mid \; & \langle expr \rangle \; '<>' \; \langle signal \rangle^* \\ \mid \; & 'switch' \; \langle branch \rangle^* \; 'end' \\ \mid \; & 'let' \; \langle ident \rangle^* \; 'in' \; \langle relation \rangle^* \; 'end' \end{aligned}$

$\langle pattern \rangle ::= \langle identifier \rangle \mid '(' \; \langle pattern \rangle \; ',' \; \langle pattern \rangle \; ')'$

$\begin{aligned} \langle expr \rangle ::= \; & \langle constant \rangle \\ \mid \; & \langle expr \rangle \; \langle binop \rangle \; \langle expr \rangle \\ \mid \; & \langle unop \rangle \; \langle expr \rangle \\ \mid \; & \langle expr \rangle \; \langle expr \rangle \\ \mid \; & 'sigrel' \; \langle pattern \rangle^* \; 'where' \; \langle relation \rangle^* \; 'end' \end{aligned}$

$\langle decl \rangle ::= 'let' \; \langle identifier \rangle \; \langle pattern \rangle^* \; '=' \; \langle expr \rangle$

$\langle module \rangle ::= \langle decl \rangle^*$

**Figure 1.** Hydra$_2$ abstract syntax.

### 2.1. Functional Hybrid Modelling

Functional Hybrid Modelling (FHM) is inspired by Functional Reactive Programming (FRP) [17], in particular by one of its implementations, Yampa [18]. FRP is an approach to causal reactive programming, although it can also be understood as an approach to causal modelling in general. Yampa is centred around the idea of signals and signal functions. A signal is, conceptually, a time-varying quantity of a given type, while a signal function is a (pure) function on signals. In many ways, a signal function is similar to a block in programming languages like Simulink [3]. However, signal functions in Yampa are first-class values: they can be manipulated and computed like regular values in the program.

This allows for a great level of expressivity, including the possibility of expressing structural changes by allowing signal functions to change over time.

FHM differs from FRP by instead abstracting over signal relations. While in a causal modelling environment, an equation $x = y$ is directed ($y$ being used to compute $x$) in FHM and other non-causal modelling languages, this equation is a relation: a priori, nothing prevents it from being used for computing either $x$ in terms of $y$, or $y$ in terms of $x$; it is the responsibility of the implementation to make that choice. Like in FRP, signal relations are first-class values. This, as we will demonstrate below, allows for great modularity.

FHM is therefore a two-level language: the functional level, which manipulates signal relations or ordinary time-invariant values; and the signal level, where relations between signals are defined. Because of this staged nature, previous implementations of FHM were realised as embeddings in the functional programming language Haskell [19]. The capabilities of the host language was used to handle the functional level, leaving only the implementation of the signal level to be realised. Hydra$_2$ has been realised with a standalone compiler handling both phases. This was done to allow tight control over the written code and to avoid relying on the Haskell run-time system during simulation. However, what is described here could to a large extent be done in an embedded implementation too.

### 2.2. Non-Causal Modelling

Consider the electrical circuit in Figure 2. It consists of 3 components and relates 6 time-varying quantities, or *signals*: $U$, $i$, $u_r$, $i_r$, $u_c$ and $i_c$. These signals are related through equations derived from the laws of electrical circuits: for instance, $i_r$ and $u_r$ are related through Ohm's law $u_r = ri_r$ and $u_c$ and $i_c$ are related through the equation $i_c = cu'_c$.

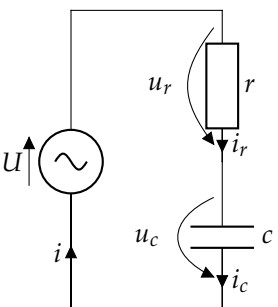

**Figure 2.** A simple RC circuit.

The behaviour of each of these components can be abstracted as a signal relation. For instance, a resistor can be modelled as follow:

```
let  resistor  r = sigrel u, i where
  u = r * i
end
```

*resistor* is a function of type $\mathbb{R} \to \mathbf{SR}\ (\mathbb{R}, \mathbb{R})$. Upon application to a concrete value for the resistance, the resulting signal relation therefore relates two real signals, $u$ and $i$, that we call its *interface variables*. The relation is simply Ohm's law that was stated earlier. As we are interested in time-varying quantity, Hydra$_2$ also has a **der** operator for taking the derivative of a signal. This allows for modelling a capacitor with the following signal relation:

```
let  capacitor  c = sigrel u, i where
  i = c * der u
end
```

To model the full circuit, these relations now have to be combined in some way. In this case, we must express that the resistor and the capacitor are connected in series. We could write a specific relation for the connected components, however recall that, in electrical circuits, the way two components connected in series behave is mostly independent

from the specific behaviour of the components. In particular, by Kirchhoff's laws, the current flowing through the two components is the same and the voltage through the two components is the sum of the voltages going through each component. As signal relations are first-class, they can be passed as arguments to functions and, therefore, one can define a signal relation for two components being connected in series like so:

```
−− serial  : SR (ℝ, ℝ) → SR (ℝ, ℝ) → SR (ℝ, ℝ)
let  serial  m₁ m₂ = sigrel u, i where
   let  u₁, i₁, u₂, i₂ in
      m₁ ◇ u₁, i₁
      m₂ ◇ u₂, i₂
      u = u₁ + u₂
      i = i₁
      i₁ = i₂
   end
end
```

The **let** block introduces local signals $u_1$, $i_1$, $u_2$ and $i_2$, respectively the voltages and current flowing through $m_1$ and $m_2$. The ◇ corresponds to signal relation application. The construction $m_1 \diamond u_1, i_1$ states that $u_1$ and $i_1$ are related through $m_1$. For instance, if $m_1$ was instantiated to *capacitor c*, this would ultimately mean that $u_1$ and $i_1$ are related through the equation $i_1 = cu_1'$.

The model from Figure 2 can be built simply using the relations for the components above and serial. Connecting it to source of alternative current yields the following relation:

```
let  circuit  = sigrel () where
   let  u, i in
      u, i ◇ serial (capacitor  c) (resistor  r)
      u = u₀ ∗ sin(2 ∗ pi ∗ f ∗ time)
   end
end
```

The result of simulating the above model is given on Figure 3, with the parameters choosen $r = 10$, $f = 1$ and $c = 0.05$.

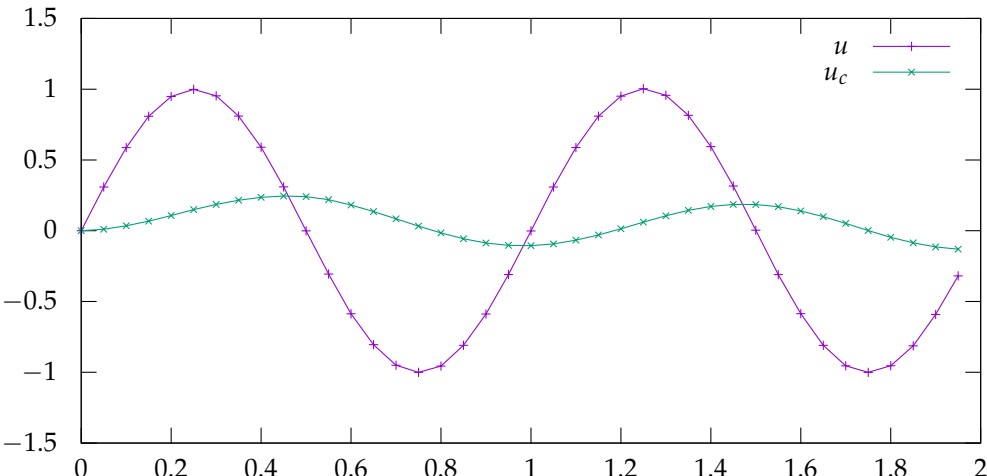

**Figure 3.** Simulation results for the *circuit* model. The runtime of the simulation was 3.0 ms (see Section 7 for a description of the hardware used) as measured by the bench tool [20].

### 2.3. Hybrid Models

A hybrid model is a model that exhibits both continuous and discrete behaviours. In the context of Hydra$_2$, it means that the behaviour of the system being simulated is allowed to change at discrete points in time. This can be useful to model some components whose

behaviour would otherwise be hard to express in a purely continuous setting, or whose resulting model would prove hard to simulate, without it being of interest to the modeller. Consider for instance an electrical diode: ideally, it is a component that only allows current to flow through it one way. If current flows in the right direction, it behaves like a wire, and otherwise like an open switch. This can be expressed in Hydra$_2$ with a switch construct as follows:

**let** *ic_diode* = **sigrel** *u, i* **where**
  **switch init** *Closed*
    **mode** *Closed* → *u* = 0
      **when up**(−*i*) → *Open*
    **mode** *Open* → *i* = 0
      **when up**(*u*) → *Closed*
  **end**
**end**

A switch is a relation made of a collection of modes. Only one mode is active at a given point in time, meaning that only the relations it specifies hold at that point in time. The mode active at the start of the simulation is given by **init**. The diode above has two modes: in the *Closed* mode, the diode acts as an wire and thus its voltage is 0, in the *Open* mode, the diode acts as an open switch, there is no constraint on the voltage but the current through the diode is 0. Each mode specifies a (possibly empty) list of transitions to other modes—a transition is made of an event specification and a target mode, when the event happens, the active mode for the switch becomes the active mode. The events are specified using special purpose combinators that produce events based on the behaviour of signals. In the case of the diode, the diode switches to the open mode when the current flowing through it becomes negative. An event is triggered on that condition using **up**, which produces an event when the signal its given crossed 0 going from a positive to a negative value.

Thanks to its support for highly structurally dynamic models, Hydra allows this approach to modelling ideal diodes to be used to model circuits involving more than one diode, such as a full wave rectifier, fully compositionally [10]. As the number of potential modes are two to the power of the number of diodes, any implementation relying on pre-enumerating all possible modes will quickly run into trouble.

### 3. Simulating a Non-Causal Model

This section explains the basics of non-causal simulation and the challenges this poses for modular compilation. The setting of Hydra$_2$ is used to make the discussion concrete. However, the described techniques and problems are relevant generally.

Simulating a Hydra$_2$ signal relation consists of finding an approximation for the behaviour over time of the signals that appear in it. Hydra$_2$ signal relations translate naturally to, eventually partial, Differential Algebraic Equations (DAE). Recovering a DAE from a signal relation can be done by gathering all the equations and local variables that appear in its body, as well as expanding other signal relations that are applied in its body. The process, also referred to as flattening, consists of duplicating the equations contained in the signal relation's body and replacing all interface variables by the signals they correspond to at the point the relation is applied. In the presence of structural dynamism, the DAE can be recovered from the set of modes every switch relation is in. For instance, the signal relation for *circuit* from Section 2.2 expands to the following DAE:

$$u_r = r i_r \tag{1}$$
$$i_c = c u'_c \tag{2}$$
$$i = i_r \tag{3}$$
$$i_r = i_c \tag{4}$$
$$u = u_r + u_c \tag{5}$$
$$u = u_0 \sin(2\pi f t) \tag{6}$$

Equation (1) (resp. (2)) come from the application of *resistor* (resp. *capacitor*). The next Equations (3)–(5) come from the application of *serial* and Equation (6) from *circuit* itself.

The simulation of a DAE is, in general, not a straightforward task: it relies on the possibility to compute, for every signal in the DAE, the value of its highest-order derivative (HOD) appearing in the equation in terms of its lower-order variables. For the above example, the HOD of all signals are their zeroth-derivative, except for $u_c$ whose HOD is $u_c'$. For convenience, we will denote by $d_x$ the order of the HOD of variable $x$ (which will then be denoted by $x^{(d_x)}$).

Computing the HODs can be done either explicitly, by symbolically manipulating the DAE into a sequence of directed, scheduled Ordinary Differential Equations (ODE) [21], or by leaving the system in implicit form and using a suitable numerical method to compute the values of the HODs from it. The first approach, known as causalisation, can yield more efficient simulation code; however, it also involves a fair amount of symbolic manipulation, in particular in the presence of non-linear equations or algebraic loops. Therefore, this is not the approach that has been chosen for Hydra$_2$. Instead, we rely on a DAE solver that is able to solve the system directly in implicit form. The difficulties associated with causalisation are discussed in Sections 8 and 9.

Both techniques require that the system is not structurally singular, that is, if the DAE is given by the residual $F = 0$, then the Jacobian matrix $\left( \frac{\partial F}{\partial x^{(d_x)}} \right)$ is not singular. Intuitively, this requirement is equivalent to saying that there must be a one-to-one mapping between HODs and equations, such that an HOD is mapped to an equation it appears in. The equation can then be thought of as solving for that variable. Unfortunately, many DAEs do not have that property directly. Consider a slight modification of the system above where, instead of being interested in how the above circuit behaves under the given tension, one wishes to know what voltage source to use so that the voltage through the capacitor, $u_c$, follows a given function $f(t)$. This is equivalent to removing Equation (6) from the DAE above and replacing it with Equation (12) in the DAE below:

$$u_r = ri_r \tag{7}$$
$$i_c = cu_c' \tag{8}$$
$$i = i_r \tag{9}$$
$$i_r = i_c \tag{10}$$
$$u = u_r + u_c \tag{11}$$
$$u_c = f(t) \tag{12}$$

The last equation cannot be used to solve for any variable, since no HOD appears in it. This equation is a *constraint* equation: it cannot be used directly to compute a solution, rather it restricts the set of solutions of the DAE. However, by doing some symbolic manipulation (e.g., replacing $u_c$ by $f(t)$ in Equation (8)), one can see that the system is solvable.

A general approach to this problem is to eliminate constraint equations by means of differentiation. Indeed, if $u_c = f(t)$ holds at every point in time, then $u_c' = f'(t)$ holds too. Replacing Equation (12) by this new, latent, equation yields a DAE which is not structurally singular. If the initial conditions of the variables in the system verify both the latent equations and the constraint equations, then the solution for the new DAE is the same as the one for the original DAE [22]. Simulation results for a Hydra$_2$ model corresponding to the system of Equations (7)–(12) is given in Figure 4, with the function $f(t) = \sin(2\pi t)$.

This process is referred to as *index-reduction*: the index refers to the number of times that parts of or the whole of the DAE must be differentiated to transform it into an ODE. A DAE that is not structurally singular is index-1. The original DAE for *circuit* is therefore index-2, since it required one differentiation to recover an index-1 DAE. Index-reduction

can be performed solely from structural information about the DAE [7,8,14] and therefore requires no symbolic manipulation of the system.

However, index-reduction is not modular. It is not possible, from a partial DAE, to know how many times each equation might appear differentiated when that partial DAE is used as part of a larger one. Generating code ahead-of-time for a partial DAE is therefore difficult, because one may have to go back to the definition of the DAE to generate code for latent equations. This is particularly problematic for hybrid systems, where the set of latent equations can change during runtime. For that reason, hybrid modelling languages usually resort to interpretation [11] or just-in-time (JIT) compilation [12,13]. Although recent work [14] has been proposed to address this issue for hybrid models, this analysis must be performed on a fully assembled model with a finite number of modes, which excludes some languages [13].

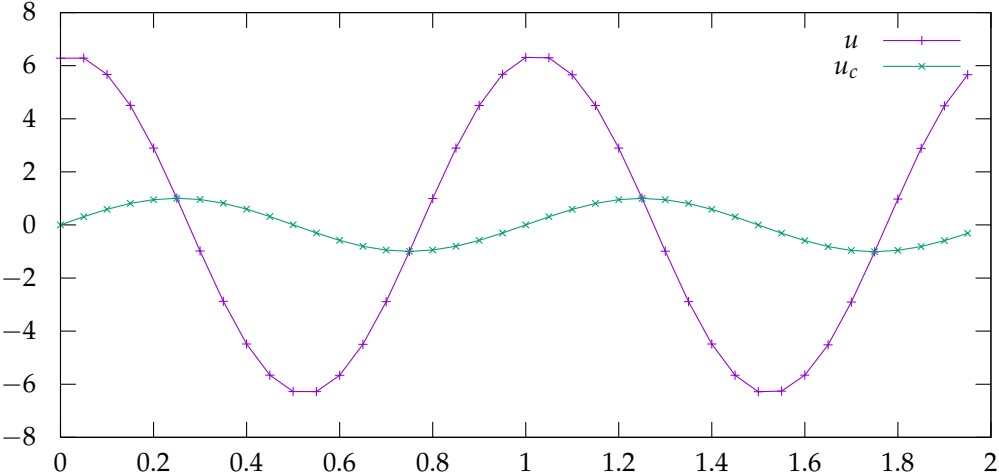

**Figure 4.** Simulation results for the model consisting of Equations (7)–(12) with $f(t) = \sin(2\pi t)$. The runtime of the simulation was 3.0 ms (see Section 7 for a description of the hardware used) as measured by the bench tool [20]. There is no significant difference in runtime with the simulation in Figure 3. A rough analysis using Linux's perf tool seems to suggest that for such small simulations, the differential equation solver itself is the limiting factor.

## 4. Higher-Order Differentiation

To counter the problem exposed in the previous section, we explore an alternative to the common ways of generating code for latent equations. Typically, first-order differentiation is applied repeatedly on the original system of equations. This is performed after index-reduction, when the model has been fully assembled. In a hybrid setting, index-reduction is performed after every mode change and may identify different sets of equations that have to be differentiated, and therefore new code has to be generated.

Instead, we consider the possibility of generating machine code that is able to compute any derivative of an expression without the need for recompilation or symbolic manipulation.

Let's consider this simple function, defined for some constant $k \in \mathbb{R}$:

$$f(t) = t^k$$

Successively differentiating against $t$ yields:

$$f'(t) = kt^{k-1}$$
$$f''(t) = k(k-1)t^{k-2}$$
$$f^{(3)}(t) = k(k-1)(k-2)t^{k-3}$$
$$\vdots$$
$$f^{(10)}(t) = k(k-1)\cdots(k-9)t^{k-10}$$
$$\vdots$$

In general, one can show that $f^{(n)}(t)$ can be computed with the following formula (Note that, when $k$ is a natural number, this formula also gives $f^{(n)}(t) = 0$, when $n > k$, since one of the factors in the product is $k - k$):

$$f^{(n)}(t) = t^{k-n} \prod_{i=0}^{n-1} (k - i). \tag{13}$$

Many common functions admit such formulæ for their $n$-th derivatives. Table 1 gives a list for all such functions in $\mathbb{R} \to \mathbb{R}$ that have a corresponding implementation in the C math header library [23].

Some of these functions do not have a direct formula for their derivatives, but can be expressed in terms of combinations of other functions. This is the case for tan and tanh, which can be expressed as respectively $\frac{\sin}{\cos}$ and $\frac{\sinh}{\cosh}$. One can also express the $n$-th derivative of tan in terms of the $(n-1)$-th derivative of its first-order derivative $\frac{1}{\cos^2}$. This approach can be used to compute the $n$-th derivative of other functions such as the inverse trigonometric and hyperbolic functions.

Table 1 also gives ways to compute the $n$-th derivative of the product (the formula being usually referred to as the generalised Leibniz rule), quotient and exponentiation of two functions. It assumes that there exists a way to compute the $n$-th derivative of the composition of two functions, which is now discussed.

Computing the $n$-th derivative of a composition can be done using Faà di Bruno's formula. It is a generalisation of the well-known chain rule, which states that $(f \circ g)'(t) = g'(t)f'(g(t))$. The formula makes use of the notion of partitions of a natural number, which we define below.

**Definition 1** (Partitions of natural numbers). *The partitions of a natural number $n$ is the set of $n$-tuple of natural numbers $(m_1, \ldots, m_n)$ such that:*

$$1 \times m_1 + 2 \times m_2 + \times + n \times m_n = n.$$

It corresponds to the number of ways a number $n$ can be split into smaller numbers. For instance, 4 has 4 partitions $(0, 0, 0, 1)$ ($4 = 1 \times 4$), $(1, 0, 1, 0)$ ($4 = 1 \times 1 + 2 \times 1$), $(2, 1, 0, 0)$ ($4 = 2 \times 1 + 1 \times 2$), $(0, 2, 0, 0)$ ($4 = 2 \times 2$) and $(4, 0, 0, 0)$ ($4 = 4 \times 1$). We denote by $P_n$ the set of partitions of $n$ and by $|P_n|$ the number of partitions of $n$, simply called the partition number.

**Table 1.** $n$-th derivative of common functions.

| Expression | $n$-th Derivative | Comment |
|---|---|---|
| $\exp(t)$ | $\exp(t)$ | |
| $t^k$ | $t^{k-n}\prod\limits_{i=0}^{n-1}(k-i)$ | For any constant $k \in \mathbb{R}$ |
| $k^t$ | $\ln(k)^n k^t$ | For any constant constant $k \in \mathbb{R}_+^*$ |
| $p(t)^{q(t)}$ | $\left(\exp(q(t)\ln(p(t)))\right)^{(n)}$ | |
| $\sin(t)$ | $\begin{array}{ll} \sin(t) & \text{if } n = 4k \\ \cos(t) & \text{if } n = 4k+1 \\ -\sin(t) & \text{if } n = 4k+2 \\ -\cos(t) & \text{if } n = 4k+3 \end{array}$ | |
| $\cos(t)$ | $\sin^{(n+1)}(t)$ | |
| $\tan(t)$ | $\left(\frac{\sin(t)}{\cos(t)}\right)^{(n)}$ | |
| $\sinh(t)$ | $\begin{array}{ll} \sinh(t) & \text{if } n = 2k \\ \cosh(t) & \text{if } n = 2k+1 \end{array}$ | |
| $\cosh(t)$ | $\sinh^{(n+1)}(t)$ | |
| $\tanh(t)$ | $\left(\frac{\sinh(t)}{\cosh(t)}\right)^{(n)}$ | |
| $\text{asin}(t)$ | $\begin{array}{ll} \text{asin}(t) & \text{if } n = 0 \\ \left(\frac{1}{\sqrt{1-t^2}}\right)^{(n-1)} & \text{if } n > 0 \end{array}$ | |
| $\text{acos}(t)$ | $\begin{array}{ll} \text{acos}(t) & \text{if } n = 0 \\ \left(-\frac{1}{\sqrt{1-t^2}}\right)^{(n-1)} & \text{if } n > 0 \end{array}$ | |
| $\text{atan}(t)$ | $\begin{array}{ll} \text{atan}(t) & \text{if } n = 0 \\ \left(\frac{1}{1+t^2}\right)^{(n-1)} & \text{if } n > 0 \end{array}$ | |
| $\text{asinh}(t)$ | $\begin{array}{ll} \text{asinh}(t) & \text{if } n = 0 \\ \left(\frac{1}{1+t^2}\right)^{(n-1)} & \text{if } n > 0 \end{array}$ | |
| $\text{acosh}(t)$ | $\begin{array}{ll} \text{acosh}(t) & \text{if } n = 0 \\ \left(-\frac{1}{\sqrt{t^2-1}}\right)^{(n-1)} & \text{if } n > 0 \end{array}$ | |
| $\text{atanh}(t)$ | $\begin{array}{ll} \text{atanh}(t) & \text{if } n = 0 \\ \left(\frac{1}{1-t^2}\right)^{(n-1)} & \text{if } n > 0 \end{array}$ | |
| $\Gamma(t)$ | $\Gamma(t)D_n(t), \begin{cases} D_1(t) = \psi(t) \\ D_k(t) = D'_{k-1}(t) + \psi(t)D_{k-1}(t) \end{cases}$ | $\psi$ is the digamma function. Programs for computing it can be found in [24]. |
| $p(t)q(t)$ | $\sum\limits_{i=0}^{n}\binom{n}{i}p^{(i)}(t)q^{(n-i)}(t)$ | |
| $\frac{p(t)}{q(t)}$ | $\left(p(t)q(t)^{-1}\right)^{(n)}$ | |

**Theorem 1** (Faà di Bruno's formula). *The n-th derivative of the composition of two $\mathcal{C}^n(\mathbb{R})$ functions f and g is a $\mathcal{C}^n(\mathbb{R})$ function given by the following formula* [25,26]:

$$(f \circ g)^{(n)} = \sum_{i=1}^{|P_n|} \frac{n!}{m_{i1}!1!^{m_{i1}} m_{i2}!2!^{m_{i2}} \cdots m_{in}!n!^{m_{in}}} \times f^{(m_{i1}+m_{i2}+\cdots+m_{in})}(g(t)) \times \prod_{j=1}^{n} \left(g^{(j)}\right)^{m_{ij}}.$$

(Faà di Bruno's formula)

*With $m_{ij}$ denoting the j-th coefficient of the i-th partition of n.*

The formula can be put into a slightly more friendly form by introducing Bell polynomials. The partial Bell polynomials [27] for *n* and *k* are defined as:

$$B_{n,k}(x_1,\ldots,x_{n-k+1}) = \sum \frac{n!}{m_{i1}!1!^{m_{i1}} m_{i2}!2!^{m_{i2}} \cdots m_{i(n-k+1)}!(n-k+1)!^{m_{i(n-k+1)}}} \times \prod_{j=1}^{n-k+1} \left(x_j\right)^{m_{ij}}, \quad (14)$$

where the sum spans over those partitions of *n* whose coefficients sum to *k*. For those, it is easy to see that $m_{ij} = 0$ if $j > n - k + 1$.

Faà di Bruno's formula can then be rewritten as:

$$(f \circ g)^{(n)} = \sum_{i=1}^{n} f^{(k)}(g(t)) B_{n,k}\left(g', g'', \ldots, g^{(n-k+1)}\right). \quad (15)$$

## 5. Order-Parametric Differentiation: Translating Equations to Imperative Code

Although the formulæ presented in the previous can be converted into machine code, the performance of the resulting code would likely be poor. The main culprit, as we will see, is Faà di Bruno's formula. In this section, we study the cause of these problems by comparing the operations performed when using Faà di Bruno's formula to those generated by a typical compiler using repeated first-order differentiation. We then propose a scheme to mitigate these problems.

### 5.1. First-Order Forward Mode Automatic Differentiation

Automatic differentiation is a technique to efficiently compute the derivative of a function specified by a computer program. Consider for instance the following function, whose implementation in imperative pseudo-code can be found in the first column of Table 2:

$$f(t) = \sin(\underbrace{\underbrace{2t}_{x_1}}_{x_2}) + \underbrace{t^3}_{x_3}.$$

Suppose we wished to emit a program that could compute $f'$, given the value of *t*. The idea of automatic differentiation is that the program that implements this function, reduces to a composition of elementary operations and functions. Therefore, the derivative of *f* can be computed by recursively computing the derivative of each subexpression by applying the basic rules of differentiation. In contrast with symbolic differentiation, which consists of symbolically differentiating *f* and implementing the resulting function in a program, automatic differentiation allows previously computed values of the derivatives of subexpressions to be reused and, in general, only yields code that is a constant factor larger than the original code [28].

When computing higher-order derivatives, the same process can be applied to the program resulting from the first-order differentiation. Notice however that it is often beneficial to explicitly introduce additional subexpressions to make higher-order differentiation efficient.

**Table 2.** Program to compute $f$ and its derivatives.

| $f(t)$ | $f'(t)$ | $f''(t)$ |
|:---:|:---:|:---:|
| $x_1 := 2 \times t$ | $\begin{aligned} x_1 &:= 2 \times t \\ x_1' &:= 2 \end{aligned}$ | $\begin{aligned} x_1 &:= 2 \times t \\ x_1' &:= 2 \\ x_1'' &:= 0 \end{aligned}$ |
| $x_2 := \sin(x_1)$ | $\begin{aligned} x_2 &:= \sin(x_1) \\ y_1 &:= \cos(x_1) \\ x_2' &:= x_1' \times y_1 \end{aligned}$ | $\begin{aligned} x_2 &:= \sin(x_1) \\ y_1 &:= \cos(x_1) \\ y_1' &:= x_1' \times x_2 \\ y_2 &:= x_1'' \times y_1 \\ y_3 &:= x_1' \times y_1' \\ x_2' &:= y_2 + y_3 \end{aligned}$ |
| $x_3 := t^3$ | $\begin{aligned} x_3 &:= t^3 \\ x_3' &:= 3t^2 \end{aligned}$ | $\begin{aligned} x_3 &:= t^3 \\ x_3' &:= 3 \times t^2 \\ x_3'' &:= 6 \times t \end{aligned}$ |
| **ret** $x_2 + x_3$ | **ret** $x_2' + x_3'$ | **ret** $x_2'' + x_3''$ |

### 5.2. Order-Parametric Differentiation

To produce code that can produce arbitrary derivatives of an expression, instead of associating two values to every subexpression, its actual value and its derivative, our approach consists of associating an array that holds all $n$ derivatives of the subexpression, where $n$ is the order of the derivative that was requested. Indexing into the array at index $i$ will give the $i$-th derivative of the subexpression. This array is then filled with the successive values of the derivatives using code corresponding to the formulæ we presented earlier. We call this scheme order-parametric differentiation, in the sense that the code being generated is parameterised by the order of the required differentiation. Although this scheme requires a form of dynamic allocation (as the size of the arrays storing the derivatives of all the subexpressions are unknown), we will detail in Section 5.3 how, in the context of a modelling language, this can be efficiently handled.

A possible formulation of this method for $f$ is given below:

**for** $i \in [0, n]$ **do**

$$x_1[i] := \begin{cases} 2 \times t & i = 0 \\ 2 & i = 1 \\ 0 & \text{otherwise} \end{cases}$$

$$y_1[i] := \begin{cases} \sin(x_1[0]) & \text{if } i = 4k \\ \cos(x_1[0]) & \text{if } i = 4k + 1 \\ -\sin(x_1[0]) & \text{if } i = 4k + 2 \\ -\cos(x_1[0]) & \text{if } i = 4k + 3 \end{cases}$$

$$x_2[i] := \sum_{i=1}^{|P_i|} \cdots$$

$$x_3[i] := t^{3-i} \prod_{j=1}^{i} (3 - j + 1)$$

$$x_4[i] := x_2[i] + x_3[i]$$

**end for**

However, this naïve formulation has some flaws, the simplest of which lies in the computation of $x_3$. When the successive derivatives of $t^3$ are explicitly computed, it is easy to have the compiler do some optimisations, for instance constant folding the products $3 \times 2$ in the second derivative. When using abstract formulæ, it is impossible

for the compiler to perform this optimisation. Here the problem is worse, not only can the compiler not perform constant folding, but it may be hard for it to realise that it is performing redundant calculations. In this particular case, the problem can be solved by using an accumulator for the product of $3 - j - 1$, like so:

```
acc := 1
for i ∈ [0, n] do
    x₃[i] := t³⁻ⁱ × acc
    acc := acc × (3 − i)
end for
```

This problem was easy to solve. But there is another, harder one, in the computation of $x_2$. Let's expand the sums that compute $x_2[3]$ and $x_2[4]$, assuming we have already computed the information needed on the partitions of 3 and 4:

$$
\begin{aligned}
x_2[3] := \;& 1 \times y_1[1] \times x_1[1]^3 \\
&+ 3 \times y_1[2] \times x_1[1] \times x_1[2] \\
&+ 1 \times y_1[3] \times x_1[3]
\end{aligned}
\qquad
\begin{aligned}
x_2[4] := \;& 1 \times y_1[1] \times x_1[1]^4 \\
&+ 4 \times y_1[2] \times x_1[1] \times x_1[3] \\
&+ 6 \times y_1[3] \times x_1[1]^2 \times x_1[2] \\
&+ 3 \times y_1[2] \times x_1[2]^2 \\
&+ 1 \times y_1[4] \times x_1[1]
\end{aligned}
$$

Notice that there are some redundant computations when computing $x_2[3]$ after $x_2[4]$. In particular in the computation of the products of the derivatives of $x_1$: the product $x_1[1]^3$ can be used in computing $x_1[1]^4$, the product $x_1[1] \times x_1[2]$ can be used in computing $x_1[1]^2 \times x_1[2]$. Although this does not look like much for small values of $n$, the problem becomes critical for larger values.

Clearly, to generate efficient code, one must find a way to reuse the previously computed values of the products of the derivatives of $x_1$. Fortunately, this reuse pattern can be exploited by viewing Faà di Bruno's formula in terms of Bell's polynomials. Indeed, Bell polynomials follow this recursive equation [29]:

$$
B_{n,k}(x_1, \ldots, x_n) = \sum_{i=1}^{n-k+1} \binom{n-1}{i-1} x_i B_{n-i,k-1}, \tag{16}
$$

with the following base cases:

$$
\begin{aligned}
B_{0,0} &= 1 \\
B_{n,0} &= 0 & n \geq 1 \\
B_{0,k} &= 0 & k \geq 1
\end{aligned}
$$

Using this recurrence formula in Faà di Bruno's formula gives:

$$
(f \circ g)^{(n)} = \sum_{k=1}^{n} f^{(k)}(g(t)) \sum_{i=1}^{n-k+1} \binom{n-1}{i-1} g^{(i)}(t) B_{n-i,k-1}. \tag{17}
$$

Since the previous computation of the derivative of $f \circ g$ used the values of the Bell's polynomials, they can simply be stored as they are computed and reused. To compute the $n$-th first derivatives, space to store the values of the Bell polynomials must be allocated. This translates naturally to imperative code:

```
B := [[1]]
for i ∈ [0, n] do
    if i = 0 then
        x₂[0] := y₁[0]
    else
        x₂[i] := 0
```

$$\textbf{for } k \in [1, i] \textbf{ do}$$
$$B[i][k] := \sum_{j=1}^{i-k+1} \binom{n-1}{i-1} x_1[i] \times B[n-i][k-1]$$
$$x_2[i] := x_2[i] + y_1[i] \times B[i][k]$$
$$\textbf{end for}$$
$$\textbf{end if}$$
$$\textbf{end for}$$

This code performs $O(n^2)$ operations to compute the $n$-th derivative of $f \circ g$ from results obtained by computing the previous derivatives. This is to contrast with using the original formulation of Faà di Bruno's formula whose number of terms grows as $O(2^{\sqrt{n}})$. Additionally, this formulation only requires the value of the binomial coefficients up to $n$, instead of information on the partitions of $n$. Although the binomial coefficients could be computed at runtime (after all, like Bell's polynomial, they follow a simple recursive pattern), there is little reason to not precompute them at compile-time—it avoids unnecessary work during the simulation at the cost of little space in the resulting object code. By default, the compiler generates the binomial coefficients for the first 20 natural numbers. This limit is not entirely arbitrary—it corresponds to the number after which some of the coefficients in the original Faà di Bruno's formula exceed $2^{64}$. Another threshold could be 27, after which the coefficients in Faà di Bruno's formula exceed $1.8 \times 10^{308}$, the largest number that can be represented as a double-precision floating point number. If one wished to generate binomial coefficients until some exceed this threshold, one would need to generate them for the first 1030 natural numbers. Still, this would only translate to 4.2 MB of space in the object code. Of course, whether a DAE with such high-index would ever be encountered (let alone be simulated) is unlikely.

*5.3. Caching and Memory Management*

Our scheme, to be efficient, relies on the possibility to store the values of the derivatives of intermediate expressions. The amount of memory that's needed cannot be known at compile-time, since the number of time an equation might appear differentiated will only be known at runtime (or eventually at link-time). Therefore, some memory must be allocated to store the arrays of derivatives for all the subexpressions, as well as space for the values of Bell's polynomials. When the index is small, one could perform the allocation on the stack. However, since the required memory can be computed using the number of differentiations that will be performed, and given that this number is known after index-reduction, the memory can also be allocated just after index-reduction and reused for all computations of the residual of the DAE, at least until a structural change happens. In fact, since equations are not expected to run in parallel, the working memory can be shared among equations and therefore one only needs to allocate enough memory to cover the needs of the largest requirement.

## 6. Compiling Signal Relations

The previous section proposed a scheme to generate code able to compute the $n$-th derivative of an expression. We now detail how signal relations as a whole are compiled.

Recall that signal relations in Hydra$_2$ are first-class objects. They can be passed as arguments to functions or returned from them. As such, their representation in the target language should be first-class objects. This allows for the functional level of the language to be implemented as a regular functional language that manipulates signal relations as regular values. The signal relation must then simply provide ways to encode its behaviour in a way that can be used by a DAE solver and also the runtime system of the language, that is in charge of performing index-reduction and handling structural change. In the case of Hydra$_2$, the runtime system is a small C library which uses the Sundials solver suite [30] for simulation. Hydra$_2$ programs themselves are compiled to LLVM [31] code, which is then used to generate object files using the LLVM compiler. The resulting object files are then linked with a conventional linker to the runtime.

The first thing that a signal relation must provide is structural information, to be used for index-reduction. In our case, this involves computing the signature matrix which is then used with the Σ-method [8] to compute the set of latent equations in the given mode, as well as the set of HODs of the system.

Then, the values for the residuals (during the simulation, but also at initialisation) must be computed, using information from index-reduction. Since the code for the residual has been generated in a way that any derivative can be computed, there is no need to perform symbolic manipulations. The runtime system simply passes how many times each equation in the signal relation must be differentiated and storage large enough to hold all the results.

Finally, a signal relation must be able to provide the solver with signals to monitor in order to trigger events, and report when a structural change happens. This leads to representing a signal relation as a record of functions, each function performing one of the tasks above. When a signal relation is applied inside another signal relation, the code that is generated simply dereferences the function it needs from that record of function and calls it.

Signal relations should also capture some of their outside environment. For instance, in the model of the capacitor, the value of the capacitance must be captured inside the signal relation. Of course, every time one would need one of the functions defining the behaviour of *capacitor c*, one could execute that application and retrieve it. However, that is completely unnecessary: although the behaviour of a signal relation can change during the simulation, the signal relation itself, at the functional level and once the simulation starts, never does. For the same reason, a signal relation that is applied inside another one should be captured by the larger signal relation, to avoid needless recomputation. The reader can probably see a similarity between this approach and a form of inheritance in object-oriented languages.

### 6.1. Representing Modes

The modes of a signal relation form a tree. Equations form the leaves of that tree (as they do not have modes) and switches represent multi-way nodes. For that reason, modes are represented as nested algebraic data types (ADT).

The representation we have devised so far resembles that of objects. Thus, one might propose that modes be represented as state encapsulated in the signal relation's representation. The method in charge of handling events simply modifies that state. However, this causes subtle problems when it comes to sharing. Consider the following signal relation:

```
let foo sr = sigrel x, y where
  sr ⋄ x
  sr ⋄ y
end
```

If the state is encapsulated within an object representing *sr*, it is therefore shared among both instances of *sr*: modifying one will modify the other silently and would seriously complicate execution. Suppose that *foo* is called with the following relation:

```
let bar = sigrel x where
  switch init A
    mode A −> x = ...
      when up(x) −> B
    mode B −> x = ...
      when up(x) −> A
  end
end
```

And suppose that, at some point during execution, the event **up**($x$) triggers. The first relation in *foo*, $sr \diamond x$ will change the state of *bar* to $B$, and the second will change it back to $A$.

For that reason, signal relations should not be in charge of any state, unlike objects. Rather, the user of the signal relation should pass a representation of the state for the method of the signal relations to use and pass around accordingly. Instead of modifying an internal state, the method handling mode changes can simply construct a new version of that state to be used for the rest of the simulation (Or, but that is equivalent, mutate the current state owned by the caller. The problem is not so much mutation but rather the extent of it on the simulation.).

*6.2. Interfacing with a Solver: Deciding on a Calling Convention*

In general-purpose programming languages, functions are compiled with a particular calling convention that states how the caller passes arguments to the function and how the callee retrieves them. When compiling the methods for a signal relation, a similar choice must be made as to how information on signals is retrieved.

This is made more complicated by the nature of our language. In general purpose languages, local variables are truly local: they do not exist before entering the function and exist outside of it only if they are returned by the function. However, in a non-causal modelling language, local signals are not truly local: they live with the solver and their declaration in a signal relation is only a way to manipulate them and refer to the value the solver has guessed for them.

One must then decide on a mapping, from a collection of signals that the solver provides, to the individual signals declared in the signal relations. We consider a simple interface with the solver where the inputs are passed in as an array of arrays—the first array contains the derivatives of the first signal, the second array the derivatives of the second signal, and so forth. Deciding on the index of a given signal in that input array can be done syntactically. The generated code keeps count of the number of allocated variables. **let** blocks are compiled in such a way that the variables they declare are allocated just after the last allocated variables. Upon returning, the signal relation informs its caller of what the new count of allocated variables is. This technique can be viewed as a form of stack, like in general purpose languages, where variables are only pushed to but never popped from. This scheme allows for keeping track of where variables are at a very minimal cost (a pointer increment when a new variable is declared), without having to inspect the code. It also works naturally in a hybrid setting.

**7. Performance Evaluation**

In this section, we present benchmarks that compare the runtime of computing the residual of various equations and their derivatives, either using an explicit representation (obtained by applying first-order automatic differentiation repeatedly) or an implicit representation, as presented in Section 5. Overall, the benchmarks show that using an implicit representation result in performance ranging from on-par with the explicit form to considerably worse. However, we have some leads on improving the situation in the most problematic cases—see Section 9.1. Also, recall that our scheme may trade some performance during the simulation, but enables modular compilation, making it attractive as an alternative to just-in-time compilation. Further, nothing rules out using a combination of approaches, leveraging their respective advantages: our approach should thus be seen as complementary to existing approaches, with its own distinct characteristics profile, not necessarily as an alternative.

The benchmarks were obtained by generating LLVM [31] code from the Hydra$_2$ compiler (The resulting LLVM code as well as the benchmarking code can be found at https://gitlab.com/chupin/hydra-v2/-/tree/separate_compilation/examples/benchmark, accessed on 29 March 2021). The resulting code was then compiled with clang on the O3 optimisation level and benchmarked using Google's benchmark library (https://github.

com/google/benchmark, accessed on 29 March 2021). The benchmarks were run on a PC with a 4-core Intel i3-7100T @ 3.4 GHz and 8 GB of RAM. Results are presented in Figures 5–9. On the graph, labelled "Implicit" is the curve giving the runtime for the code generated in implicit form, meaning it can compute any $n$ derivatives. The curve labelled "Explicit" corresponds to the runtime for code that has been specifically generated by using repeated application of first-order automatic differentiation, as presented in Section 5.1. The "Slowdown" curve corresponds to the ratio of the time taken by the implicit form over the time taken by the explicit form, it should be read on the second $y$-axis.

The derivative of a product (Figure 6), when expressed using Leibniz's rule offers performance characteristics that are very close to the explicit code. For high-order differentiation, it even runs faster. The reason for that behaviour is unclear to us—one possibility is that the code becomes so large that it is detrimental to performance, due to cache effects; another possibility is that the compiler fails to performs some optimisation (possibly due to the strange arithmetic of floating-point numbers) which result in more operations overall.

The implementation of Faà di Bruno's formula does not seem to benefit from such an effect. This can viewed on the benchmarks in Figure 5, for $\exp(x)$ and Figure 8 for $x^2$. The case of exponential is a particularly good example of one problem with our approach. Since exponential is its own derivative, when the explicit code is generated, it creates many opportunities to use already computed results. These opportunities can easily be identified and exploited by the backend. Exploiting these opportunities in implicit form is much harder for the backend and that is the reason of this large gap in performance on that particular benchmark. By contrast, the benchmark for $x^2$ shows that, although the implicit form is slightly slower, the gap remains tolerable, especially as the number of differentiations rise. For $x^2$, whose derivatives do not repeat in the same way, there are indeed a lot less opportunities for the backend to generate better quality code.

The benchmarks in Figures 7 and 9 show that the computation of the successive derivatives of division is much slower in implicit form compared to the explicit form. This is due to the fact that in the implicit, form, the $n$-th derivative of $\frac{x}{y}$ is computed as the $n$-th derivative of $xy^{-1}$. In the explicit form, the usual formula $\frac{x'y-xy'}{x^2}$ is used instead. Although the two formulation are equivalent mathematically, from a computer's perspective they are not, as one generates calls to an exponentiation function while the other simply uses divisions. Analysis of the benchmarks using Linux's perf tool show that, indeed, calls to exponentiation is a significant part of the time spent computing the successive derivatives of the quotient. Note however, that the absolute runtime is not much worse than the runtime for computing the derivatives of $x^2$ (Figure 8), which is computed in a very similar way. This seems to confirm our interpretation that the difference in runtimes can be explained by the fact that the explicit case simply uses a better formula. This is of course problematic, not only for expressions in which divisions appear, but also expressions involving functions whose first-order derivative involves a division, like the inverse trigonometric and hyperbolic functions (such as asin, see Figure 9). However, alternative ways of computing the $n$-th derivative of the division that do not rely on exponentiation could provide a way to improve this situation [32].

On some benchmarks, for instance in the benchmark for $x^2$ (Figure 8), the performance gap between the implicit and explicit forms is larger for a smaller number of differentiations. This could be mitigated by using a hybrid approach, where code is generated in explicit form for a small number of differentiations and then falls back to using order-parametric differentiation for larger numbers, where the gap is smaller.

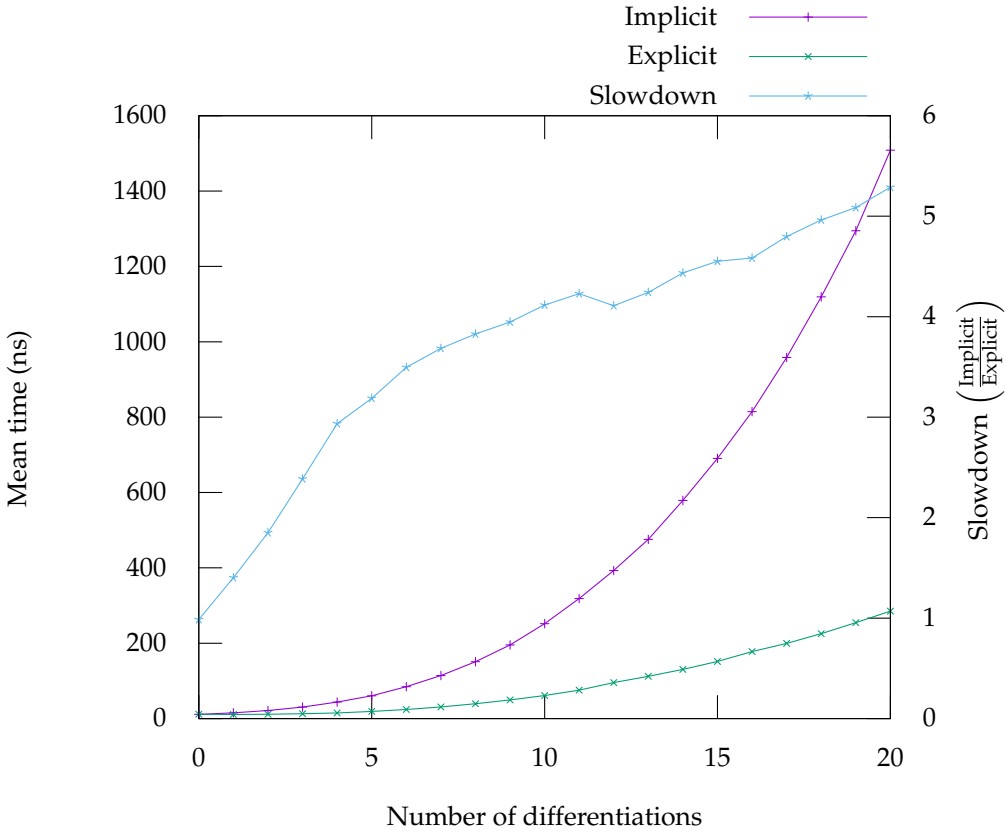

**Figure 5.** Benchmarks results for the computation of the first *n*-th temporal derivatives of exp(*x*).

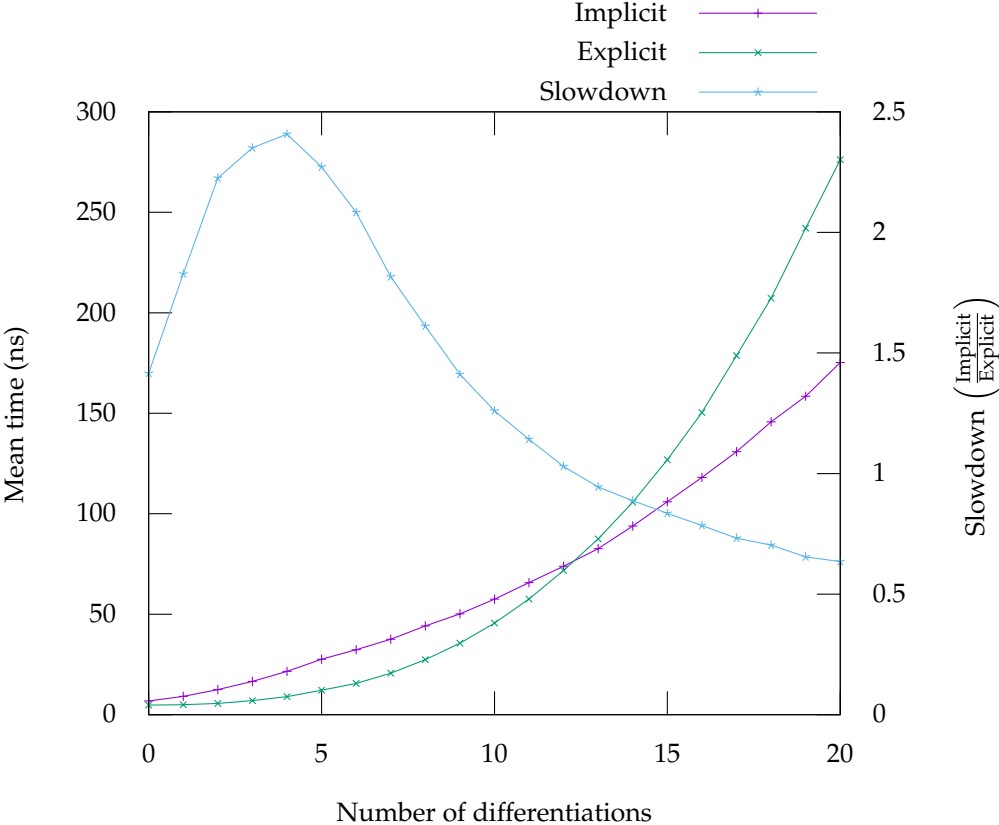

**Figure 6.** Benchmarks results for the computation of the first *n*-th temporal derivatives of *x* × *y*.

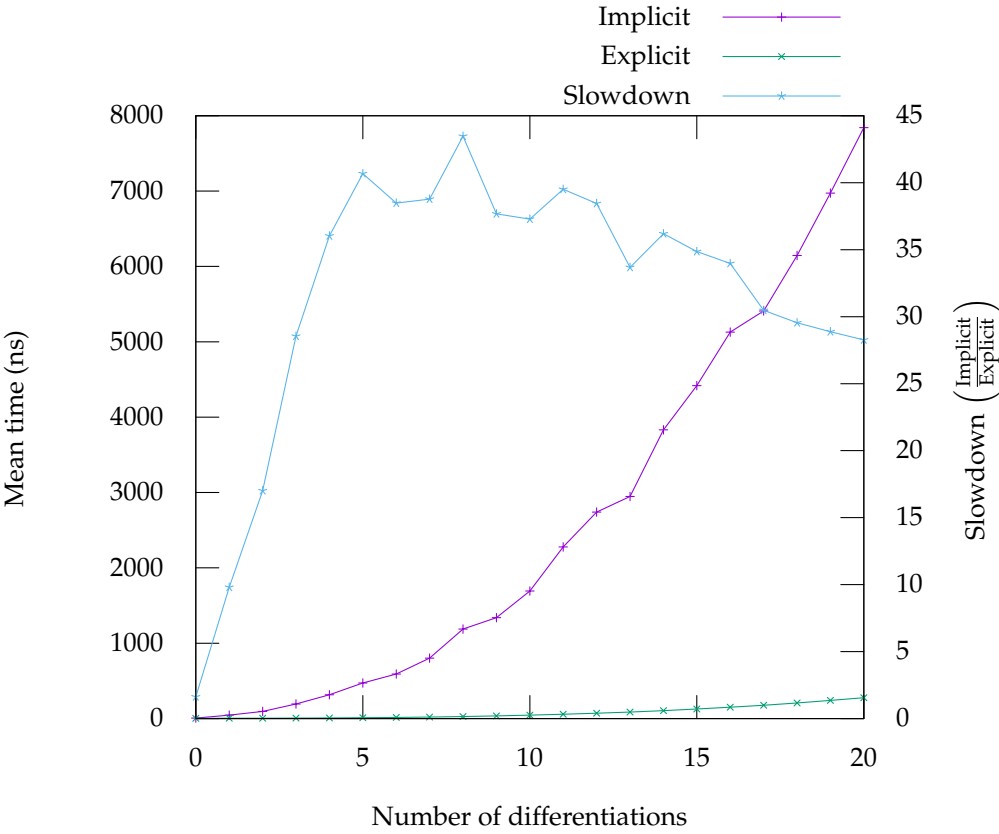

**Figure 7.** Benchmarks results for the computation of the first *n*-th temporal derivatives of $\frac{x}{y}$.

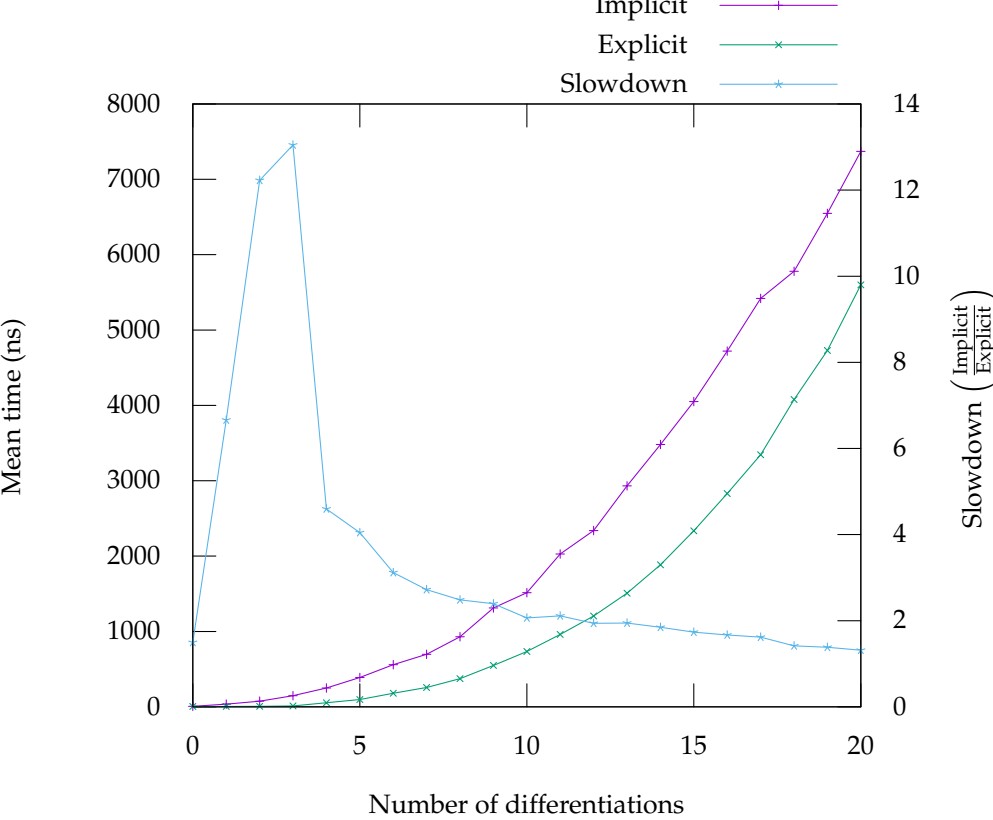

**Figure 8.** Benchmarks results for the computation of the first *n*-th temporal derivatives of $x^2$.

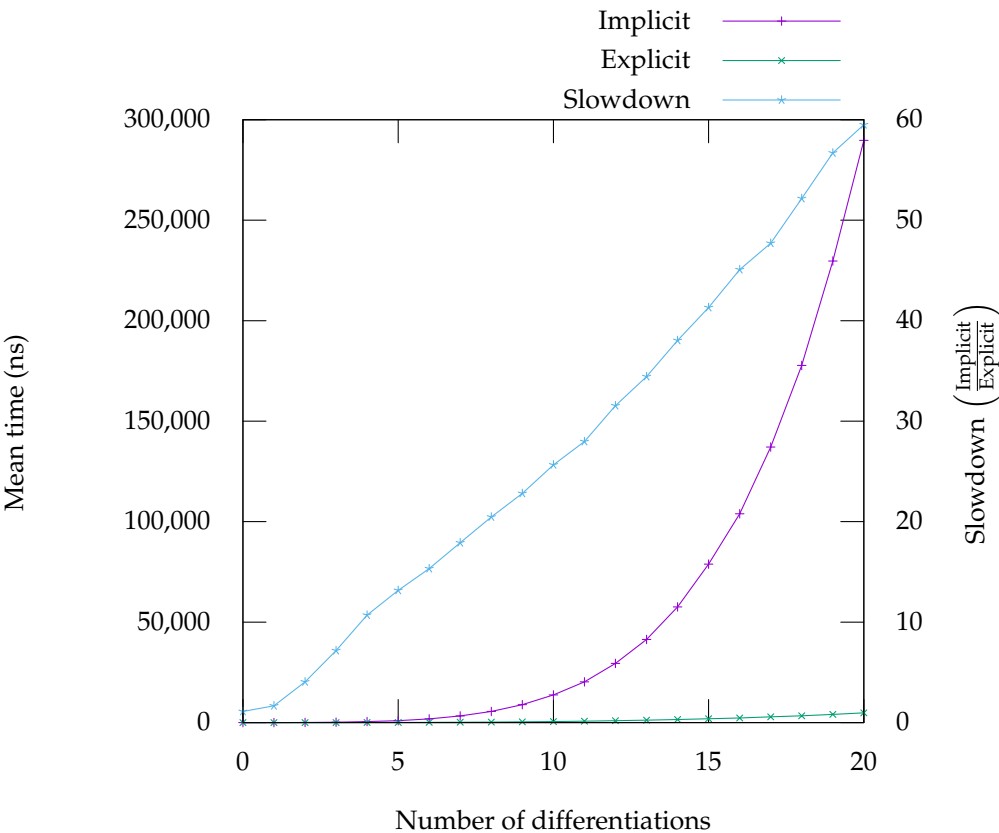

**Figure 9.** Benchmarks results for the computation of the first *n*-th temporal derivatives of asin($x$).

## 8. Related Work

The issue of modular compilation of non-causal models has been studied by a few different authors, in particular in the context of the Modelica language [1].

In [33,34], the authors explore extensions to the Modelica language that allow for separately compiling some partial models. The models that can be compiled in this way are complete, except for the definition of some signals that are considered inputs to the model. This allows for the model to be integrated as black-boxes in a causal programming environment (in this work's case, Scicos [35]). Although their work is not concerned with the separate compilation of arbitrary models, the authors note that it allows for a form of modular compilation, since two models compiled in this way can be composed in the host environment without the need for recompilation.

In [36], Zimmer presents theoretical work towards module-preserving compilation of Modelica models. The goal of the author is, when compiling a large complete model, to find ways to reuse previously generated code segments. The author focuses particularly on the issue of causalisation, which we have not considered. Central to his idea for reusing code is the notion of a causal entity, which corresponds to a particular use of a submodel with a given causality. If the same causal entity appears several time throughout a larger model, code only needs to be generated once and can then be reused by all entities. This work also contains reflections on how to decide whether to reuse the code for an entity or to regenerate it. The difficulties caused by index-reduction are mentioned but the precise way in which the author's approach is reconciled with these difficulties is left as future work. Although the goal of the author is not ahead of time compilation of partial models, the possibility that this technique could be used to distribute pre-compiled code is mentioned but the details are left for future work. It would be interesting to explore whether one could use this work in conjunction with our approach for handling latent equations to obtain a modular compiler for code in causal form.

The closest work to ours is the work by Höger in [37–39]. In particular, in [37], the author explores the modular semantics of a non-causal language using automatic higher-order differentiation. The technique for performing automatic differentiation in this work is derived from [40]. The proposed target language is able to compute an arbitrary order time derivative of equations, like in our setting, although the approach also allows for the computation of the partial derivatives of expressions, which are useful in computing the Jacobian matrix used for simulation. A proof of the correction of the translation of arbitrary expression to terms in the target language is provided and the author also provides an implementation of the target language as a Java library. The work differs from ours in our focus to generate machine code from the partial models. Our understanding is that the target language sits at a higher-level, with terms of the language being evaluated in the host language. Regardless, it would be interesting to study whether some of the shortcomings of our implementation can be solved by the approach proposed in this work. Performance is unlikely to be improved however, as the computation of composition and multiplication takes exponential time in the number of differentiations. To mitigate that problem, the author proposes to fallback onto more efficient formulations when they apply, like Faà di Bruno's formula or Leibniz's formula when they apply, which our work directly makes use of.

## 9. Limitations and Future Work

### 9.1. Performance Improvements

There are cases where the performance of the code using order-parametric differentiation could be improved. We showed for instance that computing the $n$-th derivative of the quotient with our technique performed particularly poorly. We already mentioned that one solution could be to use a specialised formulation [32] that does not rely on exponentiation. Other functions could benefit from similar formulæ, like tangent [41], exponential or the power functions [42]. Using these special rules has the downside of requiring more work from the compiler implementer, instead of relying on the rule for composition.

In cases where the performance gap between the code using repeated first-order differentiation and the code order-parametric differentiation is larger for a small number of differentiations, one solution could be to generate specialised code for the low-order derivatives using repeated first-order differentiation, and fall back to the implicit formula if higher-order derivatives are required. Deciding when, and up to how many derivatives, to generate code could be done via some heuristic (e.g., by considering the size of the code that would result from generating the code, or by considering whether the implicit form is known to have poor performance) or by user annotations. One should, in general, be careful not to generate too much code, to avoid adverse effects on caching, especially if the code is not necessarily being used.

### 9.2. External Functions

User-defined signal functions are not supported by our implementation. Currently, all functions that operate on signals (*sin*, *exp*, etc.) are built-in operators. This restriction is not due to difficulties that would occur with having to compute the $n$-th derivative of a user-defined function: simple language mechanism could be provided to allow the user to specify them (e.g., by means of annotations, like in Dymola [2]). If the definitions are simple enough, automatic differentiation could also be used.

However, if user-defined were allowed to be multivariate, a suitable way to compose these multivariate functions would have to be found. Indeed, although Faà di Bruno's formula has been extended to the multivariate case [43], it is unclear whether an implementation that use it can be made efficient. We would be interested to study other approaches to higher-order automatic differentiation (e.g., [40], used in [37]) to see if they provide solutions to this problem.

Supporting user-defined signal functions could also allow for some integration of reactive causal programming, in particular coming from Functional Reactive Programming, in FHM's implementations.

### 9.3. Modular Causalisation

The code generated by our compiler is suitable for simulation with a DAE solver. Non-causal modelling language typically also perform causalisation, so that the resulting model can be simulated with an ODE solver (eventually with the assistance of a non-linear algebraic solver, in the presence of non-linear algebraic loops). This involves symbolically manipulating the set of undirected equations, so that they appear directed and scheduled.

A partial DAE can be causalised in many ways, depending on the context in which it used. In the presence of structural changes, the causality can even change during the simulation. For these reasons, modular causality, at least in the general case, seems very difficult. A simpler goal could be to generate causal versions of only some models, for example, models that have only a few ways of being made causal or whose causality does not change during simulation. These models would then be simulated with an ODE solver and all other models with a DAE solver. Scheduling between these models would then have to dynamic, in case of structural changes.

### 9.4. Unbounded Number of Modes

Previous work on Hydra [13] presented a switch combinator much more powerful than the one implemented in Hydra$_2$—the switch combinator was allowed to compute a new signal relation at runtime. This made the number of mode potentially uncountable, as they could be generated programmatically, and offers greater dynamism and flexibility for modelling. We believe our approach could still be used in such setting. The problem of efficiently representing and handling such a switch construct in the context of interpreting FRP networks in an imperative context as already been studied in [44] and we think a similar approach could be used for an extension of Hydra$_2$.

### 9.5. Optimisations in the Context of Modularly Compiled Signal Relations

We do not present here a study on how compiling models modularly affects the simulation of the resulted code. In general purpose language, not inlining a function affects optimisation, as the function is essentially a black-box. The same applies in a non-causal language. For instance, not inlining a model can prevent propagating equalities between variables. This can cause more variables and equations to be present during the simulation than necessary. The number of equality constraint between interface variables and local variables is already a heuristic used by the Hydra$_2$ compiler to inline some signal relations. We would be interested in studying its effect and developing new ones to help generate more efficient simulation, while retaining some modularity and good compilation times.

## 10. Conclusions

In this paper, we presented the implementation of a compiler for a hybrid non-causal modelling language based on the principles of Functional Hybrid Modelling. The implementation generates machine code using LLVM in a modular fashion, even for partial models, alleviating the problem caused by latent equations by using order-parametric differentiation. This allows generating code capable of computing the value of an arbitrary derivative of an equation. The performance of our proposed scheme has been evaluated and compared with that of a more common scheme that generates code for derivatives on demand. Although the evaluation shows mixed results, we think the core idea is sound. The cost in additional simulation time is balanced by shorter compilation times, which could yield an overall simulation performance gain in the setting of hybrid languages where just-in-time compilation otherwise would have to be carried out during simulation. Further, order-parametric differentiation could be integrated with other methods, such as conventional automatic differentiation or just-in-time compilation, allowing different

techniques to be used in different parts of a model depending on their specific performance characteristics. In other words, we believe order-parametric differentiation to be an interesting complement to existing implementation techniques for cases where modular compilation is a key concern.

**Author Contributions:** Conceptualization, G.C.; Formal analysis, G.C.; Investigation, G.C.; Supervision, H.N.; Writing—original draft, G.C.; Writing—review & editing, G.C. and H.N. All authors have read and agreed to the published version of the manuscript.

**Funding:** This research received no external funding.

**Data Availability Statement:** The data generated and analyzed for this paper are available at https://gitlab.com/chupin/hydra-v2/-/tree/separate_compilation/examples/benchmark.

**Acknowledgments:** The first author would like to thank Andy McKeown for moral support and proofreading of the draft of this paper.

**Conflicts of Interest:** The authors declare no conflict of interest.

## Abbreviations

The following abbreviations are used in this manuscript:

| | |
|---|---|
| DAE | Differential Algebraic Equation |
| ODE | Ordinary Differential Equation |
| HOD | Highest-order Derivative |
| FHM | Functional Hybrid Modelling |
| FRP | Functional Reactive Programming |
| JIT | Just-in-time compilation |

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
