# Peer review of "Modular Compilation for a Hybrid Non-Causal Modelling Language"

_electronics, doi:10.3390/electronics10070814_

Round 1

Reviewer 1 Report

The authors introduced a technique they called order parametric differentiation to allow truly modular compilation. Their idea was to generate (machine) code that can compute derivatives of any order of an expression as needed, thus allowing for the ahead-of-time modular compilation of a hybrid non-causal language. They also developed a compilation scheme that enables using partial models as first-class objects in a seamless way and simulating them without the need for just-in-time compilation, even in the presence of structural dynamism. There were serval minor problems:

  1. The code and data should be uploaded onto GitHub or other publicly available websites. So, the readers can reproduce the results and validate or apply them.
  2. The equations should be numbers and be explained.
  3. Line 167 to 182 did not seem to be correctly displayed. Please check it carefully.
  4. The authors should add a case study from the real world and demonstrate the usage of the proposed model.

Author Response

1. The code and data should be uploaded onto GitHub or other publicly available
websites. So, the readers can reproduce the results and validate or apply them.

A link to a public Gitlab repository has been included on page 3 as well as on
page 16 for the benchmarks.

2. The equations should be numbers and be explained.

We've added numbering to all equations. We have also added citations to the
relevant literature for some equations whose origin was not clear from the text
alone. This include the expression of Faà di Bruno's formula in terms of Bell
polynomials (for which we now cite Bell's original work on the subject, equation
14, line 311) and the recurrence relation for Bell's polynomials (line 360).

3. Line 167 to 182 did not seem to be correctly displayed. Please check it
carefully.

Indeed, the code here has been simplified and fixed (it is now on line 176).

4. The authors should add a case study from the real world and demonstrate the
usage of the proposed model.

Unfortunately, this is not feasible within the one week turn-around for the
revision. We would also like to emphasize that the purpose here is not to
propose a replacement for existing techniques, but to suggest a complementary
technique applicable in cases when a truly modular approach (compilation to
machine code of individual model components in isolation once and for all) is
desirable, as opposed to e.g. just-in-time compilation. As such, we think that
the extensive benchmarking that we have undertaken suffices to demonstrate the
feasibility of the approach, and to allow implementors of non-causal modelling
languages to make an informed choice when our proposed technique might be
applicable.

In relation to the main concerns of reviewer #1 and #3, we have clarified that
we see our technique as a complement to existing techniques, not as a
replacement. We have tried to make this point clearer through modifications in
the abstract (line 12-14), introduction (line 81-85) and conclusion (lines
686-688).

Reviewer 2 Report

This work tackles the design and implementation of a compiler. Most of the paper is devoted to the implement differential equations of different orders.  

This has little to do with electronics except for a minor example shown. No simulations are done with electronic circuits and there is no comparison of such outputs with other known simulators and/or measurements to verify the effectiveness in electronics design. 

In my opinion this paper, while having merit, does not fit in a journal on electronics and is of limited interest or use to the general audience of this journal. 

Author Response

1. In my opinion this paper, while having merit, does not fit in a journal on
electronics and is of limited interest or use to the general audience of this
journal.

We are plased to note that reviewer #2 finds that our contribution has "merit"
and is very happy with almost all aspects of the paper as such. The main
objection is wheteher the paper is in scope for a journal on electroncis. We
note that our paper specifically is a contribution to a special issue on "Tools
and Languages for Object-Oriented Modeling and Simulation". We believe our
contribution is highly relevant in that context, and that the appropriateness
for publishing in a journal on electronics follow from the relevance of this
class of langauges to the field of electronics as such, while the immediate
application of our proposed technieques to serious simulation in the field of
electronics is not someting that we set out to explore in the present paper.
That said, in earlier work, we have demonstrated the relevance of better support
for structural dynamism in the field of electronics:

   Henrik Nilsson and George Giorgidze. Exploiting structural dynamism in
   Functional Hybrid Modelling for simulation of ideal diodes. In Proceedings
   of the 7th EUROSIM Congress on Modelling and Simulation, Prague, Czech
   Republic, September 2010. Czech Technical University Publishing House.

Two paragraphs discussing that work have been added on lines 48-52 and lines
213-217.

Reviewer 3 Report

The author introduced an order parametric differentiation method to enable the modular compilation of non-causal modelling. In general, the paper is well written by providing detailed description of the research background and the proposed method. However, I have the following concerns about the manuscript:

  1. Although the proposed method can fulfill the modular compilation of non-causal modelling, the performance evaluation demonstrates significantly poorer performance than the explicit methods. So I’m wondering whether the proposed method could really achieve any performance improvement when solving a real problem. Therefore, the authors should come up with a specific problem to demonstrate the superiority of the proposed method.
  2. The method is proposed mainly to enable modular compilation. However, there is no performance comparison with any of the existing modular compilation methods.
  3. “fundamentals” in line 88 should be “fundamental”.
  4. There is an extra “control” in line 117.

Author Response

1. Although the proposed method can fulfill the modular compilation of non-causal
modelling, the performance evaluation demonstrates significantly poorer
performance than the explicit methods. So I’m wondering whether the proposed
method could really achieve any performance improvement when solving a real
problem. Therefore, the authors should come up with a specific problem to
demonstrate the superiority of the proposed method.

We view the proposed technique as a complement to existing techniques, not as a
replacement. And we are certainly not arguing that it is superior to existing
techniques in cases where existing techniques are adequate. Rather, we believe
we have demonstrated that our technique is a feasible approach (compilation to
machine code of individual model components in isolation once and for all) for
cases where true modular compilation is a key requirements, for example for
allowing for the design and implementation of non-causal modelling languages
with better support for highly structurally dynamic systems.

In relation to the main concerns of reviewer #1 and #3, we have clarified that
we see our techinique as a complement to existing techniques, not as a
replacement. We have tried to make this point clearer through modifications in
the abstract (line 12-14), introduction (line 81-85) and conclusion (lines
686-688).

2. The method is proposed mainly to enable modular compilation. However, there is
no performance comparison with any of the existing modular compilation
methods.

Our understanding is that the only method that allows for modular compilation in
a way similar to ours is the work by Höger. We have added a paragraph in our
litterature review to explain why we don't think the proposed technique could
perform better than ours (line 593-599).

3-4. “fundamentals” in line 88 should be “fundamental”. There is an extra
“control” in line 117.

This has been fixed.

Round 2

Reviewer 1 Report

The authors have answered my questions.

Author Response

We thank the reviewer for the feedback in the first round and are happy to see we have now answered their questions.

Reviewer 2 Report

This work tackles the design and implementation of a compiler. Most of the paper is devoted to the implement differential equations of different orders.  

Very little has been done to address my concerns.

Still, no simulations are presented with electronic circuits and there is still no comparison of such outputs with other known simulators and/or measurements to verify the effectiveness in electronics design. 

Reviewer 3 Report

The authors have revised the manuscript correspondingly.  I would recommend for the acceptance.

Author Response

We thank the reviewer for the feedback in the first round and are happy to see we have answered their questions.